# Personalized Assessment of Normal Tissue Radiosensitivity via Transcriptome Response to Photon, Proton and Carbon Irradiation in Patient-Derived Human Intestinal Organoids

**DOI:** 10.3390/cancers12020469

**Published:** 2020-02-18

**Authors:** Ali Nowrouzi, Mathieu G. Sertorio, Mahdi Akbarpour, Maximillian Knoll, Damir Krunic, Matthew Kuhar, Christian Schwager, Stephan Brons, Jürgen Debus, Susanne I. Wells, James M. Wells, Amir Abdollahi

**Affiliations:** 1Heidelberg Ion-Beam Therapy Center (HIT), Department of Radiation Oncology, Heidelberg University Hospital (UKHD), 69120 Heidelberg, Germany; mahdi.akbarpour@dkfz-heidelberg.de (M.A.); m.knoll@dkfz-heidelberg.de (M.K.); C.Schwager@dkfz-heidelberg.de (C.S.); juergen.debus@med.uni-heidelberg.de (J.D.); a.amir@dkfz-heidelberg.de (A.A.); 2German Cancer Consortium (DKTK) Core Center, Clinical Cooperation Units (CCU) Translational Radiation Oncology and Radiation Oncology, National Center for Tumor Diseases (NCT), German Cancer Research Center (DKFZ), 69120 Heidelberg, Germany; 3Heidelberg Institute of Radiation Oncology (HIRO), National Center for Radiation Research in Oncology (NCRO), German Cancer Research Center (DKFZ) and Heidelberg University Hospital (UKHD), 69120 Heidelberg, Germany; Stephan.Brons@med.uni-heidelberg.de; 4Division of Molecular and Translational Radiation Oncology, Heidelberg Medical Faculty (HDMF), Heidelberg University, 69120 Heidelberg, Germany; 5Cancer and Blood Diseases Institute, Cincinnati Children’s Hospital Medical Center, Cincinnati, OH 45229, USA; Mathieu.Sertorio@cchmc.org (M.G.S.); susanne.wells@cchmc.org (S.I.W.); 6Light Microscopy Facility, German Cancer Research Center, 69120 Heidelberg, Germany; D.Krunic@dkfz-heidelberg.de; 7Division of Developmental Biology, Cincinnati Children’s Hospital Medical Center, Cincinnati, OH 45229, USA; Kuhv2r@gmail.com (M.K.); james.wells@cchmc.org (J.M.W.); 8Division of Endocrinology, Cincinnati Children’s Hospital Medical Center, Cincinnati, OH 45229, USA; 9Center for Stem Cell and Organoid Medicine, Cincinnati Children’s Hospital Medical Center, Cincinnati, OH 45229, USA

**Keywords:** human intestinal organoids (HIO), radiotherapy, gastrointestinal (GI) toxicity, particle therapy, carbon ions, personalized medicine

## Abstract

Radiation-induced normal tissue toxicity often limits the curative treatment of cancer. Moreover, normal tissue relative biological effectiveness data for high-linear energy transfer particles are urgently needed. We propose a strategy based on transcriptome analysis of patient-derived human intestinal organoids (HIO) to determine molecular surrogates for radioresponse of gastrointestinal (GI) organs at risk in a personalized manner. HIO were generated from induced pluripotent stem cells (iPSC), which were derived from skin biopsies of three patients, including two patients with FANCA deficiency as a paradigm for enhanced radiosensitivity. For the two Fanconi anemia (FA) patients (HIO-104 and 106, previously published as FA-A#1 IND-iPS1 and FA-A#2 IND-iPS3), FANCA expression was reconstituted as a prerequisite for generation of HIO via lentiviral expression of a doxycycline inducible construct. For radiosensitivity analysis, FANCA deficient and FANCA rescued as well as wtHIO were sham treated or irradiated with 4Gy photon, proton or carbon ions at HIT, respectively. Immunofluorescence staining of HIO for 53BP1-foci was performed 1 h post IR and gene expression analyses was performed 12 and 48 h post IR. 53BP1-foci numbers and size correlated with the higher RBE of carbon ions. A FANCA dependent differential gene expression in response to radiation was found (*p* < 0.01, ANOVA; n = 1071 12 h; n = 1100 48 h). Pathways associated with FA and DNA-damage repair i.e., transcriptional coupled nucleotide excision repair, homology-directed repair and translational synthesis were found to be differentially regulated in FANCA deficient HIO. Next, differential regulated genes were investigated as a function of radiation quality (RQ, *p* < 0.05, ANOVA; n = 742 12 h; n = 553 48 h). Interestingly, a gradual increase or decrease of gene expression was found to correlate with the three main qualities, from photon to proton and carbon irradiation. Clustering separated high-linear energy transfer irradiation with carbons from proton and photon irradiation. Genes associated with dual incision steps of TC-NER were differentially regulated in photon vs. proton and carbon irradiation. Consequently, SUMO3, ALC1, POLE4, PCBP4, MUTYH expression correlated with the higher RBE of carbon ions. An interaction between the two studied parameters FA and RQ was identified (*p* < 0.01, 2-way ANOVA n = 476). A comparison of genes regulated as a function of FA, RQ and RBE suggest a role for p53 interacting genes BRD7, EWSR1, FBXO11, FBXW8, HMGB1, MAGED2, PCBP4, and RPS27 as modulators of FA in response to radiation. This proof of concept study demonstrates that patient tailored evaluation of GI response to radiation is feasible via generation of HIO and comparative transcriptome profiling. This methodology can now be further explored for a personalized assessment of GI radiosensitivity and RBE estimation.

## 1. Introduction

Radiosensitivity of gastrointestinal (GI) organs such as the esophagus, duodenum, and rectum, are critical, e.g., for the curative treatment of thoracic tumors, and pancreatic or prostate cancer. Moreover, stomach and small bowel toxicity is a major limitation for high dose palliative radiotherapy on lumbar bone metastasis or retroperitoneal lymph nodes. Empirical data on normal tissue complication probability (NTCP) considering radiation dose and volume such as QUANTEC (quantitative analysis of normal tissue effects in the clinic) has substantially improved radiotherapy [1]. However, these approaches do not consider the broad spectrum of inter-individual heterogeneity in GI response to radiation that may result from genetic predisposition of the patient. Ionizing radiation (IR)-induced gastrointestinal (GI) toxicity imposes a challenge for the radiotherapy of cancer patients [2,3,4]. These toxic effects are classified as acute, developing within days or weeks of radiation exposure, or as chronic, developing months and years after treatment [5]. Acute effects related to direct DNA damage, p53 activation, cell cycle arrest, senescence and apoptosis most likely also lead to the development of chronic effects such as the loss of the epithelial integrity, activation of inflammatory processes which may further promote damage to the vascular and induced connective tissue fibrosis, atrophy and chronic tissue dysfunction. This underlines the importance of investigating early and late effects of radiation toxicity coordinately [5].

For understanding the causal connection of acute and late effects of IR-induced GI toxicity most ideally, experiments have to be performed in a tissue specific context which makes animal experiments indispensable at present. Such animal experiments are difficult to design and perform since long observation times and large animal numbers are required. With the engineering of human pluripotent-stem-cell-derived gastrointestinal organoids [6,7,8] a highly promising technology is available which can enable the rapid monitoring of personalized acute and chronic effects of radiation on the GI tract in vitro. Protocols for establishing organoids from pluripotent cells have been reported for many organs (intestine, stomach, pancreas, brain, liver, eyes) [9,10]. Organoids consist of many types of cells which differentiate from progenitor stages during organoid development and also have shown to have the ability to recapitulate normal tissue functions [10,11,12,13]. Thus, organoids resemble human organs and, therefore, constitute an attractive technology to complement in vivo animal studies for understanding mechanism involved in human tissue repair, regeneration and therapy response/toxicity in an authentic context [9,10].

Normal tissue is aimed to be spared by high-precision irradiation with particles. However, compared to conventional photon irradiation, uncertainties of relative biological effectiveness (RBE) of protons, and even more so for heavier ions with higher linear energy transfer (LET) such as carbon radiotherapy exist [14,15]. We here have evaluated the feasibility of human intestinal organoids (HIO) to recapitulate patient radio-sensitivity. Moreover, with the advent of high-precision irradiation using protons and heavier ions such as carbon ions at HIT [14,16], we sought to evaluate the feasibility of this approach to decipher the transcriptome response as a surrogate for RBE of different radiation qualities (RQ). To evaluate whether potential differences in the transcriptional response to radiation is influenced by the genetic predisposition of the patients we have compared radiation-associated gene expression patterns in normal to FANCA deficient HIO which were derived from iPSC of individuals with FA [17]. The sensitivity of FA patients to chemotherapy agents involved in DNA damaging processes such as cisplatin and mitomycin C is well known. However, although adjuvant radiation therapy is recommended for high stage tumors, systematic identification of safe radiation therapy (RT) dose levels for FA cancer patients who receive RT have not been reported [18]. As a proof of concept study, this work demonstrates the feasibility of individual assessment of radio-sensitivity by ex-vivo irradiation of HIO combined with a series of surrogates of radiation response including transcriptome and radiation-induced nuclear foci (RIF) formation.

## 2. Results

### 2.1. Detection and Quantification of Radiation-Induced Foci (RIF) in Pluripotent Derived Human Intestinal Organoids

Human intestinal organoids (HIO) were generated from the H1 embryonic stem cell line (H1-HIO) and two independent (104-HIO and 106-HIO) previously described induced pluripotent stem cell (iPSC) lines [17]. The 104- and 106-HIO were generated from patient specific iPSC which were derived from keratinocytes (104-HIO) or fibroblasts (106-HIO) biopsies of Fanconi anemia patients carrying a mutation in the FANCA gene. The H1-HIO was derived from the H1 embryonic stem cell line cells and is considered as the normal healthy phenotype (Figure 1a). FANCA expression is required for the efficient generation of iPSC [19,20] and HIO [17]. Therefore, prior to reprogramming, iPSC patient biopsies had been transduced with lentivirus expressing the wild type FANCA gene under doxycycline (DOX)-inducible control to rescue the FANCA deficiency. The established 104- and 106-HIO maintain characteristics of FA deficient cells in response to cytotoxic agents [17] in the absence of DOX-induced complementation with FANCA. In the following we will refer to HIO in which (DOX)-induced expression of FANCA expression was induced by doxycylin (DOX) as FANCA rescued and in the absence of DOX as FANCA-deficient HIO.

From each line single organoids were embedded in Matrigel in 96-wells and subjected to three different radiation qualities (Figure 1b). HIOs derived from all three pluripotent lines were irradiated with 4Gy physical doses of photon as well as clinical mixed beam field (SOBP) irradiation with proton or carbon ions (Figure 1c). A calculation of the local effect model (LEM) was performed predicting RBE doses of proton and carbon compared to photon, respectively (Appendix A). One hour after radiation exposure, immunostaining was performed on HIO cryosections using an antibody against 53BP1 which accumulates at radiation-induced DNA-double-strand breaks (DSB) forming radiation-induced foci (RIF) (Figure 1d). Foci numbers and size were imaged by fluorescence microscopy and quantified with in house developed ImageJ macros (see methods). Compared to the non-irradiated control we detected a higher 53BP1-foci number per cell in carbon compared to proton irradiated HIO from both the FANCA proficient and deficient background (Figure 1d). Moreover, we determined the average size of 53BP1-foci in wt and FA-deficient HIO which were irradiated with proton and carbon beams respectively. This analysis shows that as previously described irradiation with carbon ions results in larger 53BP1-foci as compared to photon and proton beams which is considered an indicative for more complex DNA damage patterns induced by high-LET irradiation (Figure 1d).

### 2.2. Comparative Gene Expression Analyses between HIO Derived from Healthy Individuals with Fanconi Anemia in Response to Radiation

For analyzing the transcriptional response of HIO to ionizing radiation, 3 organoids on average were embedded in Matrigel in 96-wells. One 96-well plate served as the non-irradiated control and three 96-well plates were irradiated with 4Gy photon, proton and carbon irradiation separately (Appendix A). Gene expression analyses were performed 12 and 48 h after irradiation. First, we confirmed that the 104-HIO and 106-HIO expressed higher levels of FANCA in the presence of doxycycline (Appendix A). Next we determined whether baseline gene expression differences can be observed in HIOs with a functional and non-functional Fanconi anemia pathway. Therefore, the expression level of genes in 104-HIO FANCA rescued vs. FANCA-deficient and 106-HIO FANCA rescued vs. FANCA-deficient were compared (Appendix A). Both in the 104-HIO and 106-HIO only minor gene expression differences could be detected between the presence and absence of the FANCA gene which could not be linked to the Fanconi anemia pathway directly (Appendix A). In comparison to the H1-HIO, gene expression differences were predominately detectable between the 104-HIO obtained from iPSCs derived from keratinocyte whereas the gene expression pattern of the 106-HIO obtained from iPSCs derived from fibroblast was more similar to the H1-HIO (Appendix A). For the comparison of gene expression differences between FA proficient and deficient HIO only the 106-HIO was used. Interestingly in response to irradiation which acts as cytotoxic agent inducing DNA double strand breaks (DSB), a comparison between the H1-HIO and the 106-HIO revealed that in the presence of a functionally active FANCA gene (106-HIO FANCA rescued) the global DNA repair and metabolism-associated transcriptional response programs are more similar to the H1-HIO. In contrast, the FANCA-deficient 106-HIO (106-HIO FANCA-deficient) showed a differential transcriptional response (Figure 2a,b, Appendix A). A comparison of gene expression values between healthy HIO and FA-derived 106-HIO (106-HIO FANCA rescued and 106-HIO FANCA-deficient) identified genes (*p* < 0.01, ANOVA; n = 1071 12h; n = 1100 48h) which showed a differential regulation in response to irradiation (Appendix A). The 106-HIO FANCA rescued gene expression signature was more similar to H1-HIO were as in 106-HIO FANCA-deficient harbored greater differences as shown by hierarchical cluster (Manhattan distance) analysis (Figure 2a,b, Appendix A). From these genes 660 genes showed a significant (*p* < 0.01, *T*-test) gene expression difference after 12 h and 789 genes after 48 h in 106-HIO FANCA-deficient compared to the H1-HIO (Figure 2a,b; Appendix A). The top functional categories which were significantly affected by differential gene regulation in irradiated FANCA-deficient 106-HIO FANCA-deficient after 12 h were processing of pre-mRNA and rRNA (*p* < 0.006; CSNK1E, DDX49, EXOSC7, NOC4L, NOP14, PWP2, RPL27A, RPL9, RPS14, RPS15A, RPS25, RPS4X, RPS9, SIK1, WDR18, WDR46) and dual incision in transcription coupled nucleotide excision repair (*p* < 0.02; ERCC1, ERCC2, GTF2H4, POLD2, POLE2, POLR2J, POLR2K, PRPF19). After 48 h genes regulating FOXO-mediated transcription (*p* < 0.001; BCL6, CCNG2, CDKN1A, CREBBP, NPY, RBL2, YWHAB) and targets of SUMO E3 ligases (*p* < 0.02; AAAS, CREBBP, DNMT3B, NCOR2, NR1I2, NR2C1, NSMCE1, NUP153, NUP160, NUP37, NUP54, PIAS2, SMC3, WRN, ZBED1) were specifically differentially regulated in FA-deficient HIO.

#### FANCA-Deficient HIO Show a Differential Gene Expression Signature of DNA Damage Response Relevant Genes Compared to FANCA Proficient HIO

A comparison of 106-HIO FANCA rescued and 106-HIO FANCA-deficient identified 224 DDR-associated genes which were differentially (*p* < 0.05, *T*-test) regulated in the 106-FANCA-deficient HIO (Appendix A). From these genes 50% were also identified to distinguish the H1-HIO from the 106-HIO FANCA-deficient (*p* < 0.05, ANOVA). Generally, both after 12 and 48 h post exposure to IR core components of the transcription coupled nucleotide excision repair machinery (TC-NER) as well as genes involved in the regulation of translational synthesis (TLS) and homology-directed repair showed differential regulation specific for 106-HIO FANCA-deficient (Figure 3a; Appendix A). After 12 h the strongest genes which were specifically upregulated in 106-HIO FANCA-deficient were SFN, POLD4, CDKN1A, TRIM25, and HIST1H2BE (Figure 3b). The strongest downregulated genes were dominated by genes which specifically regulate the transition form G1-G1/S phase (CDC25B, CDK6, MCM2, CDK4, CDKN1B, MCM7) and separation of sister chromatids at the mitotic spindle checkpoint (PTTG1, CENPF, RCC2, AURKB, CKAP5) (Figure 3b). Further we detected a differential regulation of Fanconi anemia-related genes (*p* < 10–13; ATRIP, BRCA1 EME1, ERCC1, FANCB, FANCC, FANCD2, FANCE, FANCG, RPA1, RPS27A, UBA52, UBE2T) and translesion synthesis regulating genes (*p* < 10–16; CUL4A, MAD2L2, POLD1, POLD2, POLD3, POLD4, POLE2, RFC3, RFC5, RPA1, RPS27A, UBA52, UBA7) in 106-HIO FANCA-deficient (Appendix A). Interestingly all except for EME1 and POLD4 were downregulated (Appendix A) in 106-HIO FANCA-deficient. After 48 h the most significant genes which were specifically upregulated in 106-HIO FANCA-deficient were CDKN2A, CDKN1A, YWHAE which are related to the development of cellular senescence. The top downregulated genes were HIST1H2BD, HIST1H4H, SUMO2, and RAD51AP1 involved in regulating homology-directed repair by homologous recombination or single strand annealing. Further FOS a component of the SMAD3/SMAD4/JUN/FOS complex and APEX1 a AP endonuclease class 1 enzyme involved in DNA base excision repair (BER) pathway of DNA lesions induced by oxidative and alkylating agents were specifically downregulated in 106-HIO FANCA-deficient. SKP2 and MCM2 both involved in the switching of replication origins to a post-replicative state were also specifically downregulated in 106-HIO FANCA-deficient (Figure 3b).

### 2.3. Photon, Proton and Carbon Radiation Induce Common and Radiation Quality Specific Gene Expression Signatures in Irradiated HIOs

A Pearson correlation on normalized gene expression data revealed that independent of the radiation quality used, the most significant gene expression differences were detectable between 12 and 48 h in all three HIO tested (Figure 3a,b and Appendix B and Appendix C). Hierarchical clustering resulted in two main cluster branches which were composed of distinct gene expression regulation patterns after 12 and 48 h post IR (Appendix A). A summary of common transcriptional signatures of HIO in response to photon, proton and carbon irradiation is presented in the Appendix B and Appendix C.

Interestingly following a Pearson correlation we observed transcriptional differences between photon and particle (proton and carbon) irradiated HIO indicating differences in acute and late response of HIO exposed to photon compared to proton and carbon beams (Appendix A). This sub-clustering was more pronounced 12 h compared to 48 h post irradiation were the transcriptional response of HIO to carbon beams showed a unique and separate pattern compared to photon and proton beams after 48 h (Appendix A). After 12 h 742 genes (Appendix A) were differentially regulated (*p* < 0.05, ANOVA) between photon compared to proton and carbon beams (Appendix A). Among the top functional categories affected by differentially regulated genes were genes involved in dual incision steps of transcription coupled nucleotide excision repair (TCEA1, ERCC5, POLR2B, ERCC3, POLR2D, UBC, MCRS1, XAB2, REM1, POLR2I, DDB1). After 48 h the transcriptional response of photon and proton beams was more similar. A total of 553 genes (Appendix A) could be identified which were differentially regulated (*p* < 0.05, ANOVA) between photon and proton compared to carbon beams. Carbon ion beams have a higher LET and superior RBE in comparison to photon and proton ions were as the RBE of Proton is similar to Photon (Appendix A). Therefore, we investigated whether specific gene expression signatures correlate and are unique in response to carbon irradiation by performing pavlidis template matching (PTM) analyses on gene expression data of the 12 and 48 h time points. After 12 h we found 230 transcriptional signatures which were significantly (*p* < 0.01) correlated with the higher RBE of carbon ions and 225 which were anti-correlated (Figure 4a; Appendix A). Among the correlated genes cellular processes regulated by sterol regulatory element-binding transcription factor (LSS, FASN and FDPS) were overrepresented. Expression of SUMO3 a small ubiquitin-related modifier which accumulates at DSB and in interaction with the SUMO E3 ligases PIAS1 and PIAS4 regulates DNA repair of DSB [21] correlated with higher RBE of carbon ions. Further the DNA helicase CHD1L also known as ALC1 [22] and the DNA polymerase POLE4 [23] also correlated to higher RBE of Carbon ions. PCBP4, which is activated by p53, and which suppresses cell proliferation by inducing apoptosis and cell cycle arrest [24,25] correlates with the higher RBE of Carbon ions as well as the protein translation regulating genes (RPL34, RPS15, RPS28, MRPS34, MTRF1L, MRRF, ERAL1, KARS, SARS2). Among the top functional categories which were anti-correlating with a higher RBE of carbon beams were P53 and FOXO regulated cell death genes (BCL6, CNOT8, EP300, FAS, FOXO3, GADD45A, JMY, JUN, L3MBTL, MAPKAPK5, PPP2R5C, PRKAG1, SGK, SGK1, TNFRSF10D, UBC). Genes regulating membrane trafficking USO1, PRKAG1, ARL1, COG6, FCHO2, COG1, SCARB2, SEC31A, VAMP4, VPS25, SEC22A, TRIP11, YIPF6, GCC2, AAK1 were also anti-correlated with the superior RBE of Carbon beams. After 48 h 301 transcriptional signatures were significantly (*p* < 0.01) correlated with a higher RBE of carbon beams and 343 genes were anti-correlated (Figure 4a; Appendix A). The most dominant correlated gene was the adenine DNA glycolase MUTYH which is involved oxidative DNA damage repair by signaling apoptosis after the induction of single strand breaks [26]. Other genes involved in base excision repair by translational synthesis (RFC2, RFC1, HIST1H2BE, HIST1H2BC) were also correlated with carbon beams. Anti-correlated genes were the ubiquitin protein ligase FBXW7 and other members of the E2 ubiquitin-conjugation enzyme family (UBE2E1, CTR9, HIP2, UBE2H, SELS).

### 2.4. Identifying the Role of the FA Pathway in Response to Different Radiation Qualities and Determining Molecular Surrogates of Normal GI Tissue Response in FA Patients

Since differential gene regulation patterns were identified in HIO based on the FANCA status and the applied radiation qualities we further analyzed whether the interaction of genetic predisposition, radiation quality and superior RBE of carbon beams results in distinct transcriptional signatures identifying genes which potentially could resemble surrogates for a personalized assessment of normal tissue radiosensitivity of FA patients. First a two-way-ANOVA was performed to specifically assess the interaction of the FA and radiation quality effect on irradiated HIO with no functionally active FA pathway. These analyses revealed that particular genes were differentially regulated in response to the different radiation qualities used and the FANCA expression status. A total of 476 genes (Appendix A; Appendix A) were significantly (*p* < 0.01) correlated to FANCA status linking both the FA and radiation quality effect. The Fanconi anemia pathway genes FANCB, POLN, DCLRE1A, UBA52, and FANCF genes were significantly (*p* < 0.01, *T*-test) downregulated after radiation only in FANCA-deficient HIO lines. Genes involved in the negative regulation of MET Proto-Oncogene Receptor Tyrosine Kinase signaling (LIG1, UBA52 and SH3GL2) were also specifically downregulated (*p* < 0.01, *T*-test) in response to the radiation qualities in FANCA-deficient HIO lines. In contrast NOTCH1 signaling genes (FBXW7, CUL1, UBA52, and HDAC7) were specifically upregulated (*p* < 0.01, *T*-test) in response to IR in FANCA-deficient HIO lines. Second we compared three sets of gene lists which were defined as (I) differentially regulated in 106-HIO FANCA-deficient cells compared to healthy cells, (II) differentially regulated in comparison of radiation qualities and (III) differentially regulated with higher RBE to define a core set of genes resembling potential surrogates of FA-associated normal tissue radio-sensitivity and/or -toxicity. Based on this analysis for each time point after radiation a core set of genes was defined which captures both the FA phenotype as well as the effect of different radiation qualities as well as superior RBE of carbon ions (Figure 4b). These core genes a predominantly involved in p53-regulated signaling processes (Figure 4c).

## 3. Discussion

The identification of biomarkers for accessing radiation-induced normal tissue toxicity has important implications for personalized radiation therapy. Especially in the context of pathogenic genetic predispositions of cancer patients, cell-based models are required which can potentially unmask molecular determinants of normal tissue toxicity of individual patients rapidly and efficiently. Organoids derived from tumor biopsies or from induced pluripotent stem cells comprise a promising technology to close this gap and to analyze personalized molecular determinants of radiation-induced tissue toxicity. Their multicellular organization and organ-like function provides unique advantages over 2D cell-based systems and potentially animal models.

We reasoned that analyzing gene expression signatures of organoids in response to radiation could define molecular surrogates of normal tissue toxicity. To evaluate our hypothesis human intestinal organoids were generated from iPSC and subjected to three different radiation qualities (photon, proton, carbon) at physical isodoses of 4Gy to analyze their gene expression patterns post radiation. Therefore, two different time points after radiation were chosen (12 and 48 h) to generate sequential gene expression signatures potentially resembling acute and late phases of radiation-induced toxicity. As a paradigm for the response of a highly sensitive tissue and for the testing of genetic predispositions in transcriptional radiation response we have compared the gene expression signature of HIO generated from iPSC of healthy individuals and Fanconi anemia patients carrying a mutation in the FANCA gene in response to radiation. Fanconi anemia is a rare genetic disease which is characterized by an impaired response to DNA damage and in particular a deficiency to repair DNA interstrand crosslinks (ICL). FA patients have a significantly increased risk of developing solid cancers, with a 28% cumulative incidence of solid cancers by the age of 40. The high sensitivity to chemotherapy agents (particularly cisplatin and mitomycin C) is known and, therefore, often avoided in post-operative treatments. Adjuvant radiation therapy is recommended to treat patients with high stage tumors but sensitivity to radiation therapy is not well characterized on the molecular level in FA patients to the present, with FA patients showing different level of radio-sensitivity.

In comparison to non-irradiated controls independent of the genetic background or radiation quality, common gene expression signatures of irradiated HIO could be defined by an upregulation of p53-regulated processes and a downregulation of ATR-mediated stress response 12 h post IR. Gene expression patterns after 48 h affected significantly more functional categories compared to transcriptional alterations after 12 h, showing that signaling cascades associated with the response of radiation sequentially increase in complexity after IR. The predominant molecular processes affected by gene expression changes after 48 h were associated with a downregulation of cell cycle checkpoint genes, regulation of the mitotic spindle checkpoint, separation of duplicated chromosomes such as components of the anaphase promoting complex/cyclosome (APC/C). This may indicate that mitotic cell division is delayed at the level of spindle checkpoint and kinetochore attachment [27] which is required for the separation of sister chromatids during DNA replication [28,29]. We cannot rule out that this common program is due to the exposure of the HIO to relatively high doses of 4Gy which could resemble gene signatures correlated with toxic levels of radiation. Nevertheless, these distinct transcriptional changes in response to radiation manifest a multilevel interplay between DNA damage and pathways associated with protein turnover with the aim of restoring tissue hemostasis in response to radiation-induced cellular stress. Based on the gene expression analyses it is likely that these processes are regulated at the level of p53 response, protein translation, ubiquitination and SUMOylation to control mitotic cell division by common cellular stress response mechanisms.

The FA-derived HIO under normal non-irradiated conditions showed a transcriptional phenotype comparable to HIO derived from healthy patients. Similar to mouse models of FA [30] upon exposure to genotoxic agents such as radiation FA-deficient HIO showed a transcriptional pattern which is clearly distinguishable from genetically healthy conditions. FA-deficient HIO showed a differential regulation of genes involved in processing or pre-mRNA and rRNA, dual incision steps occurring during TC-NER, translational synthesis, homology-directed repair and FA pathway genes after 12 h sequentially switching to the differential regulation of FOXO-mediated transcription and targets of SUMO E3 ligases after 48 h. Very recently the FA disease has been linked to defects in ribosomal biogenesis on the level of rRNA processing [31] which could explain why genes involved in such processes are differentially regulated in FA deficient HIO in response to radiation. The inefficient repair of DNA interstrand cross-links (ICL) is considered as the major cytogenic defect in FANCA-deficient cells [32]. The repair of ICL requires the interplay of FA, TC-NER, HR and TLS pathways [33]. Since these same pathways show differential regulation between healthy and FA-deficient HIO, this indicates that the genetic predisposition underlying FA leads to an altered disease specific transcriptional response upon exposure to radiation. The fact that compared to healthy conditions translational synthesis (TLS) polymerases involved in tolerating replication stress-induced DNA damage [34] is downregulated in FANCA-deficient HIO further shows that the presence of FANCA is required for the regulation of REV1-, POLK- and POLI-dependent TLS pathways as previously reported [35]. In addition, several FA components and FA pathway-related genes are simultaneously downregulated in FA-deficient HIO, indicating that that FA repair pathways fail to be activated in response to IR in FANCA defective versus healthy cells.

The importance of the observed differentially regulated pathways for the repair of ICL in FA-deficient HIO suggests that similar to the sensitivity of FA cells to mitomycin C and cisplatin, radiosensitivity in FA-deficient cells can also be defined as the failure to repair ICL. The effect of lesions related to ICL in relation to radiation-induced damage is controversial [36] at present. Our comparative analyses of FA-deficient and healthy HIO however show that genes relevant for the interplay of FA-, TC-NER- and TLS- pathways essential for the repair of ICL are actively transcribed in response to radiation in healthy versus FA-deficient HIO, which are expected to have impaired ICL repair. This indicates that the frequency and repair of ICL may be relevant after exposure to radiation, in particular for defining the sensitivity of the GI tract for FA patients.

In this respect genes involved in the formation of TC-NER Pre-Incision Complex and coordination of dual incision in TC-NER are among the strongest factors distinguishing the transcriptional response of photon compared to proton and carbon irradiated HIO 12 h after radiation. This transcriptional pattern suggests that the frequency of interstrand crosslinks (ICL) which requires dual incision complexes to be repaired by TC-NER differs between photon and particle beams. On the contrary clustering analyses 48 h after radiation separated photon and proton from carbon gene expression patterns indicating that at later time points high LET irradiation and higher RBE induced by carbon ions leads to a unique transcriptional response including prominent DNA repair relevant factors such as SUMO3, ALC1, POLE4, PCBP4, and MUTYH. Based on the identified gene lists a core set of genes was identified which were simultaneously differentially regulated based on the FA status, radiation qualities and higher RBE. Over 50% of these genes are involved in molecular processes regulating p53 activity and may serve as molecular surrogates resembling normal GI radio sensitivity and/or toxicity of FA patients compared to healthy patients. Further evaluation of these candidate genes is required in a larger cohort of patient-derived healthy and FA-derived HIO at different dose ranges. Based on our analyses it will be interesting to analyze DSB repair kinetics and transcriptional patterns in specific cell population to determine the sensitivity of different compartments and cell types in patient-derived organoids.

## 4. Materials and Methods

### 4.1. Generation and Cultivation of Human Intestinal Organoids

The HIO were derived from induced pluripotent stem cells as described previously [4]. Briefly, human induced pluripotent stem cells were grown in 6-well surface plates (Nunclon, ThermoFisher, Germany) coated with Matrigel (BD Biosciences, Germany) and maintained in mTESR1 media (Stem Cell Technologies, Germany). For induction of definitive endoderm (DE), iPS cells were passaged with Dispase (Invitrogen, Germany) and plated in a Matrigel-coated, Nunclon surface 6-well plate. 10 μM Y27632 compound (Sigma Aldrich, Germany) was added to the media for the first 3 days. Cells were allowed to grow until they reached 60–70% confluence. Cells were then treated with 100 ng/mL of Activin A for 3 days. DE was then treated with hindgut induction medium (RPMI 1640, 2% complemented FCS, 500 ng/mL rhFGF4 and 3 µM Chiron99021) for 4 days to induce formation of mid-hindgut spheroids. 30–50 Spheroids were then plated in ice-cold Matrigel (BD) and maintained in intestinal growth medium (Advanced DMEM/F-12, N2, B27, 15 mM HEPES, 2 mM l-glutamine, penicillin-streptomycin) supplemented with 50 ng/mL rhEGF (R&D), 100 ng/mL Noggin (R&D) and 500 ng/mL rhRSpondin1 to generate human intestinal organoids (HIOs). Media was changed at 3 days. Mature HIOs were developed approximately after 40 days. The 106 and 104 lines were cultivated in intestinal growth medium in which 100 ng/mL doxycycline was added in order to rescue the FANCA deficiency by ectopic expression of the FANCA.

### 4.2. Irradiation of Human Intestinal Organoids with Photon, Proton and Carbon Irradiation

For irradiation 2-3 HIO per well of a 96-well plate were embedded in Matrigel covered with intestinal growth medium. 96-well plates were irradiated with 4Gy photon, proton and carbon beams separately. Proton or carbon ions were administered at the Heidelberg Ion-Beam Therapy Center (HIT) as described [16,37]. For organoids which were irradiated with proton and carbon particles a 15 mm spread-out bragg peak (SOBP) with a middle depth of 120 mm was chosen. For proton beams energy ranges between 128.7–137 MeV were chosen with an average LET of 4.6 keV/µm ranging between 3.7–8.1 keV/µm in the SOBP. For carbon beams the energy range was between 2972–3152 MeV with an average LET of 86 keV/µm ranging from 65–155 keV/µm in the SOBP.

### 4.3. Immunofluorescence Staining and Microscopy of Human Intestinal Organoids

Approximately 1 h after exposure to photon, proton, and carbon irradiation the HIO which were embedded in Matrigel 96-well plates were transferred in a 15 mL Falcon tube containing 4% paraformaldehyde while removing as much of the Matrigel as possible and gently rocked for 1 h at 4 °C. HIO were washed three times with PBS at room temperature for 10 min and transferred to a 30% sucrose/PBS solution for overnight incubation at 4 °C. The HIO were then transferred to the center of an embedding mold and covered with Cryo-OCT and incubated at room temperature for 30 min. A dry ice/70% ethanol mixture was prepared, and embedded molds were placed in this mixture until the OCT turned completely white. For long-term storage the embedded molds were placed at −80 °C before cryosection was performed. For cryosection the OCT blocks were warmed up for approximately 15 min to reach a temperature of −20 °C. Sections were cut and placed on microscopy slides for subsequent immune staining of radiation-induced DSB as described (PMID: 26254681, PMID: 27494855). Briefly, Irradiated HIO, slides were washed twice with PBS and blocked 30 min with 3% BSA in PBS+0.1% Triton. The p53-binding protein 1 (53BP1) polyclonal antibody (Cell Signaling; 4937) was prepared at a dilution of 1:200 and incubated over night at 4 °C in a humid incubation chamber. Slides were washed three times in PBS and incubating with 5 µg/mL goat anti-rabbit Alexa-555 for 1 h at room temperature in the dark. Slides were washed again for 3 times with PBS and 1 time in ddH2O and mounted with DAPI mounting reagent (life technologies; P36931). Microscopy and imaging were done with Olympus R cell microscopy at the DKFZ core facility. For the quantification of 53BP1-foci numbers and area the 40× magnifications were used and the foci numbers and are counted with the ImageJ software. In parallel the are values of DAPI were quantified to determine the cell number in each image. Values for foci numbers were normalized to the DAPI area of each image to determine foci numbers and area per cell. These values were then normalized to the values obtained for 0Gy to calculate the fold change of 53BP1 numbers and size in HIO after exposure to photon, proton and carbon irradiation.

### 4.4. Isolation of RNA from Human Intestinal Organoids

HIO were frozen 12 and 48 h after irradiation in RLT buffer (Quiagen 79216) and stored at −80 °C before RNA was isolated. For RNA isolation, the HIO were homogenized and total RNA was isolated using the RNeasy Mini Kit (Qiagen, Germany) according to the manufactures protocol followed by quality control performed by Bioanalyzer and quantification by nanodrop. For genome-wide expression analyses Illumina HumanHT-12 v4 Expression BeadChip Kit was utilized and the microarray hybridization and scanning were performed at dkfz core facility.

### 4.5. Analysis of Gene Expression Values

Gene-expression analysis was performed as described [15,38] Gene transcripts with 30% or more non-assessable reads on the microarray were excluded from the analysis. Total norm intensity was performed on all samples. To further correct for background-noise from the microarray readout, only genes with a mean intensity-level of ≥100 were included in the analysis. The gene expression data of irradiated HIO were normalized to the arithmetic mean values of the respective control-groups. To identify differentially regulated genes ANOVA, two-class unpaired *t*-test, and multiple testing correction (Bonferroni) was applied using the in-house developed Software (write out first time) SUMO, version 1.61j [38]. Hierarchical clustering using Manhatton distance (average sistance) was performed with SUMO. Microarray data is available online at ArrayExpress [39].

## 5. Conclusions

This study shows that gene expression analyses of pluripotent derived human intestinal organoids proves feasible for distinguishing transcriptional programs corresponding to normal radiosensitvity modeled by HIO of healthy patients compared to highly radiation sensitive tissues modeled by HIO of FA patients. Interestingly unique gene expression signatures after exposure of HIO to photon, proton and carbon irradiation were identified which indicate a possible difference in the frequency of interstrand crosslinks induced by photon and particle beams. Combined with the identified differential gene expression signatures of three different radiation qualities, and genes correlating to increased RBE induced by high LET carbon irradiation, candidate molecular surrogates of normal tissue radiotoxicity in FA patients compared to healthy conditions have been identified. These may prove clinically relevant for determining FA patient radio- sensitivity and/or toxicity doses for photon, proton and carbon ions.

## Figures and Tables

**Figure 1 cancers-12-00469-f001:**
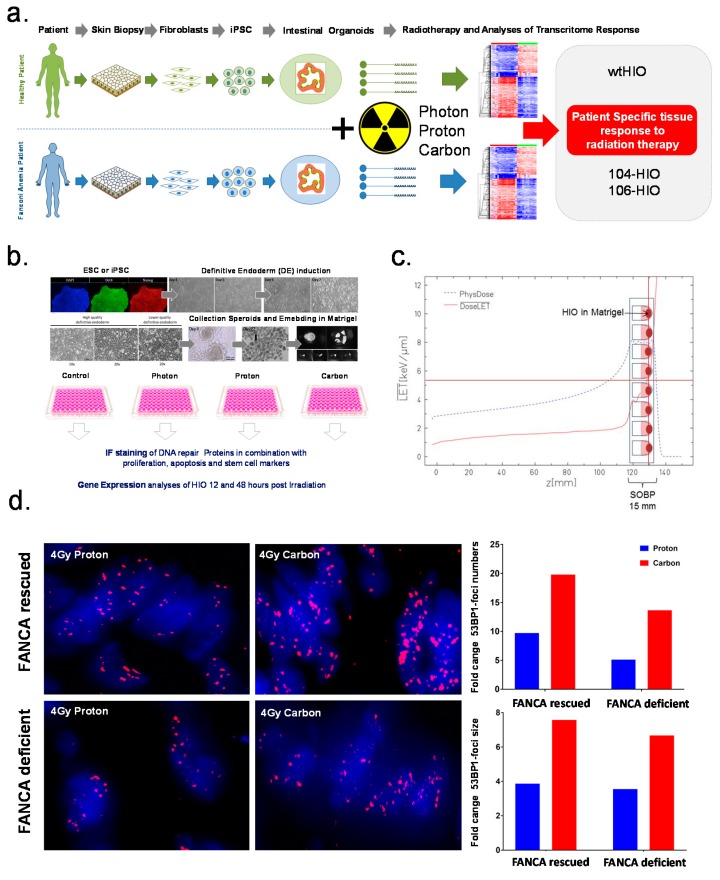
Irradiation of pluripotent derived human intestinal organoids with photon, proton and carbon irradiation. (**a**) Schematic outline on how organoids can be used to identify molecular surrogates of normal tissue radiation sensitivity and toxicity (**b**) Experimental outline of the study. Images of different stages of organoid development were kindly provided by the Center for Stem Cell and Organoid Medicine, Cincinnati. Human intestinal organoids (HIO) were generated as described previously and then embedded singularly in Matrigel in 96-plates. Following exposure to 4 Gy photon, proton and carbon irradiation HIO were cryo frozen for IF staining and stored for subsequent RNA isolation. (**c**) LEM prediction of carbon ions and positioning of organoids for irradiation. (**d**) Radiation-induced repair foci were visualized in cryo sections of irradiated HIO with IF stainings of 53BP1. The intensity of radiation-induced damage in HIO was quantified by measuring the 53BP1 numbers and size per cell. Represented values are normalized to background levels of 53BP1-foci.

**Figure 2 cancers-12-00469-f002:**
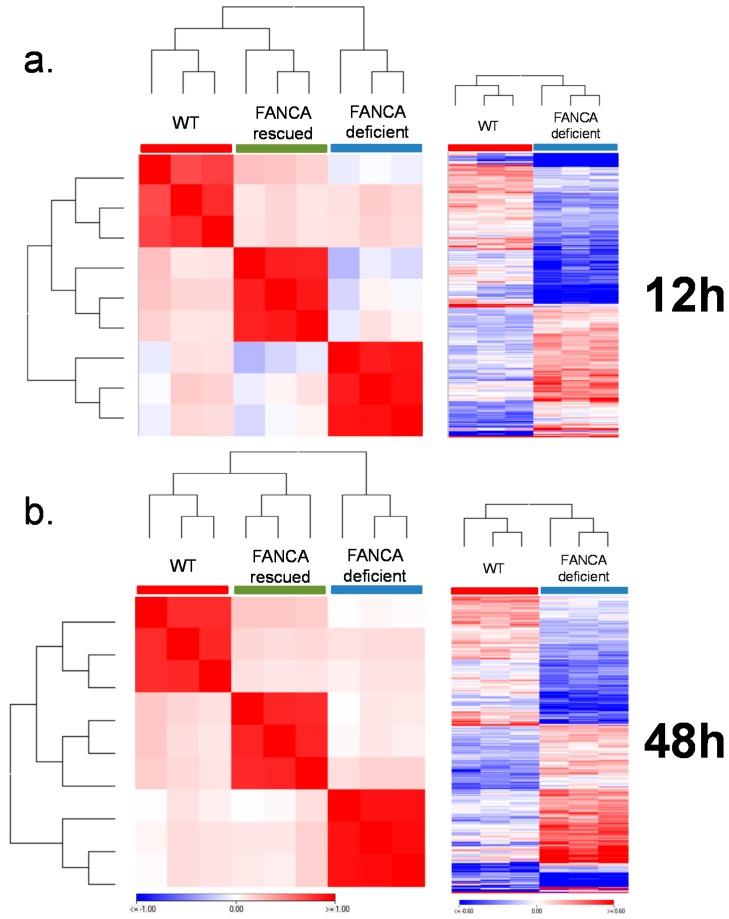
Transcriptional patterns in response to photon, proton and carbon irradiation in wild type and FANCA-deficient HIO derived from healthy and Fanconi anemia individuals. The transcriptional status of genes related to the Fanconi anemia pathway was analyzed in the HIO-106 line 12 (**a**) and 48 h (**b**) post exposure to photon, proton and carbon ions. The Pearson correlation between wt and FANCA rescued and FANCA-deficient cells derived from FA patients clearly distinguishes FANCA-deficient HIO from HIO which express intact FANCA. For each time point gene expression values between healthy HIO and FA-derived 106-HIO (FANCA-deficient and FANCA rescued) were identified (*p* < 0.01, ANOVA; n = 1071 12 h; n = 1100 48 h) which are differentially regulated. All radiation qualities were combined for this analysis.

**Figure 3 cancers-12-00469-f003:**
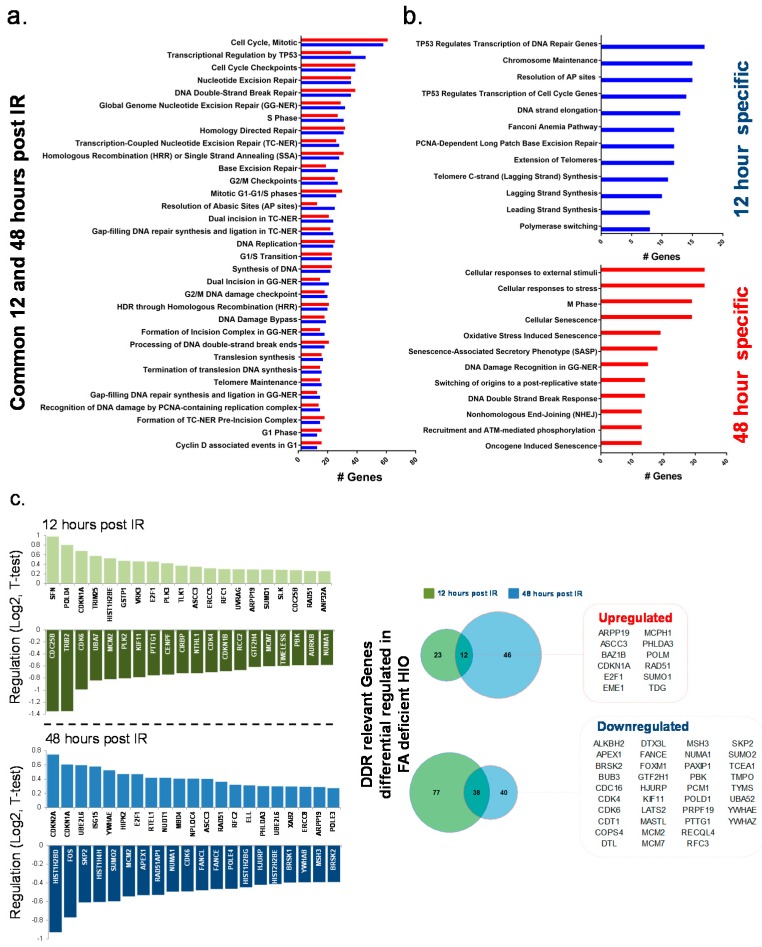
Differential expression of DNA damage response-related genes between wild type and Fanconi anemia (FA)-deficient HIO. (**a**) Represented are all the significantly enriched gene categories affected by genes which were significantly differentially regulated between wt and FA-deficient HIO both after 12 and 48 h. (**b**) The functional categories of genes which were time point specifically (12 and 48 h post IR) differentially regulated in FA-deficient cells were further divided. (**c**) Represented are the top differentially upregulated and downregulated genes for the 12 and 48 h time point separately and the gene list of genes commonly up- or downregulated is presented in the Venn diagram.

**Figure 4 cancers-12-00469-f004:**
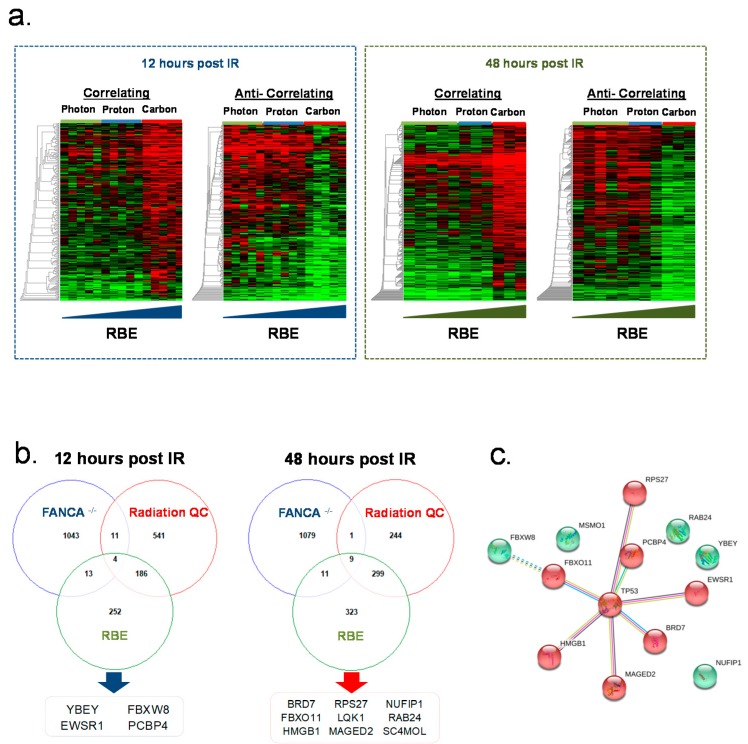
Gene expression changes as a function of Fanconi anemia, radiation quality and RBE. (**a**) gene expression heatmaps of genes which are correlated or anti-correlated with the higher RBE of carbon ions for the 12 and 48 h timepoint (*p* < 0.05, PTM). (**b**) The Venn diagram represent the overlap of genes modulated by FA, radiation qualities (RQ), and relative biological effectiveness (RBE) status for each timepoint resembling the core set of genes modulating FA response to radiation. (**c**) Genes which were modulated as a function of FA, RQ and RBE were analyzed with the STRING Search Tool for the Retrieval of Interacting Genes/Proteins. Represented are genes which interact with each other over the TP53 access.

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
