# Peer review of "Personalized Assessment of Normal Tissue Radiosensitivity via Transcriptome Response to Photon, Proton and Carbon Irradiation in Patient-Derived Human Intestinal Organoids"

_cancers, 2020, doi:10.3390/cancers12020469_

Round 1

Reviewer 1 Report

The authors present an interesting and thoroughly executed research project using elegant iPSC-generated organoid models with doxycyclin-inducable FANCA expression to study molecular radiobiology responses. The manuscript is of interest and warrants publication after the following issues have been addressed:

- FANCA ON and FANCA OFF are confusing terms, because FANCA OFF includes a double negative (the deficiency is “off”). This should be clarified, e.g. by using “FANCA deficient” and “FANCA rescued” or similar.

- In the introduction, it is unclear why the authors make this selection among GI organs: Radiosensitivity of gastrointestinal (GI) organs such as esophagus, duodenum, rectum are critical, e.g. for the curative treatment of thoracic tumors, and pancreatic or prostate cancer.” Why specifically the esophagus, duodenum and rectum? E.g., stomach and small bowel toxicity is a major limitation for high-dose palliative radiotherapie, eg on lumbar bone metastases or retroperitoneal lymph nodes.

- Introduction: “Within the DFG KFO-214” This abbreviation should be explained. Is it necessary at all?

- Fig 1a could be improved by including the cell line numbers H1-HIO and 104-HIO and 106-HIO in the healthy/FANCA rows.

- Is Table 1 necessary in the main manuscript? Perhaps the authors agree with me it is more suitable for the supplements.

- Is it necessary to include all differentially expressed genes in the results? “(p< 205 0.006; CSNK1E, DDX49, EXOSC7, NOC4L, NOP14, PWP2, RPL27A, RPL9, RPS14, RPS15A, RPS25, 206 RPS4X, RPS9, SIK1, WDR18, WDR46) and dual incision in transcription coupled nucleotide excision 207 repair (p< 0.02; ERCC1, ERCC2, GTF2H4, POLD2, POLE2, POLR2J, POLR2K, PRPF19). After 48 hours 208 genes regulating FOXO-mediated transcription (p<0.001; BCL6, CCNG2, CDKN1A, CREBBP, NPY, 209 RBL2, YWHAB) and targets of SUMO E3 ligases (p<0.02; AAAS, CREBBP, DNMT3B, NCOR2, NR1I2, 210 NR2C1, NSMCE1, NUP153, NUP160, NUP37, NUP54, PIAS2, SMC3, WRN, ZBED1)” and further as well.

- The resolution of Fig 3 needs improvement.

The manuscript seems at times unfinished and still in concept form, e.g. in the Methods: “Describe the lines better, what are they derived from pediatric patients with FANCA skin biopsies, FA pathway deficient, would not grow HIO due to DDR failure, hence lentiviral infection for stable overexpression of FANCA under dox promotor to form HIO, once HIO formed lack of FA due to cessation of Dox leads to HIO with FA phenotype.” The methods should include information on how the authors generated Dox-inducable FANCA expression in these cell lines. The conclusion now only includes some final remarks on the value and validity of the authors' novel experimental model. perhaps they can also include some concluding remarks on the differential effects they have found between photon, proton and carbon ion radiation. The high number of abbreviations in the abstract makes it borderline unreadable and some abbreviations can probably be saved for the main text. It now reads like a conference abstract.

Author Response

Please find the response in attachment.

Reviewer 2 Report

This manuscript described gene expression analyses and personalized assessment of normal tissue radiosensitivity after irradiation by photon, proton and carbon. This is very interesting and important in this field. Overall, this is a well-written manuscript with thoughtful discussion. Although there are several minor concerns, I guess this work is worthy of being accepted after minor revision.

The same sentence is repeated in introduction session (Line 91-105 and Line 108-123). Authors have to check these. Authors described the methods of irradiation with proton and carbon in Line 456-460. But authors only described “For the organoids which were irradiated with proton and carbon particles a 15mm SOBP with a middle depth of 120mm was chosen.” Authors should describe the detail of proton and carbon beams, such as dose rate and LET.

Abbreviation notation is partially incorrect. Authors should correct these.

Author Response

Please find the response in attachment.
